# Positional SHAP (PoSHAP) for Interpretation of machine learning models trained from biological sequences

Quinn Dickinson🔟, Jesse G. Meyer🔟*

Department of Biochemistry, Medical College of Wisconsin, Milwaukee, Wisconsin

* jessegmeyer@gmail.com

## Abstract

Machine learning with multi-layered artificial neural networks, also known as "deep learning," is effective for making biological predictions. However, model interpretation is challenging, especially for sequential input data used with recurrent neural network architectures. Here, we introduce a framework called "Positional SHAP" (PoSHAP) to interpret models trained from biological sequences by utilizing SHapely Additive exPlanations (SHAP) to generate positional model interpretations. We demonstrate this using three long short-term memory (LSTM) regression models that predict peptide properties, including binding affinity to major histocompatibility complexes (MHC), and collisional cross section (CCS) measured by ion mobility spectrometry. Interpretation of these models with PoSHAP reproduced MHC class I (rhesus macaque Mamu-A1*001 and human A*11:01) peptide binding motifs, reflected known properties of peptide CCS, and provided new insights into interpositional dependencies of amino acid interactions. PoSHAP should have widespread utility for interpreting a variety of models trained from biological sequences.

**Data Availability Statement:** Data and code are available from: https://github.com/jessegmeyerlab/positional-SHAP The data including all points used to create the main figures are available from zenodo https://zenodo.org/record/5711162#.YZaK-57MJ6l.

## Author summary

Machine learning enables biochemical predictions. However, the relationships learned by many algorithms are not directly interpretable. Model interpretation methods are important because they enable human comprehension of learned relationships. Methods like SHapely Additive exPlanations were developed to determine how each input alters the model prediction. However, interpretation of models trained from biological sequences remains more challenging; model interpretation often ignores ordering of inputs. Here, we train machine learning models using biological sequence data as an input to predict peptide collisional cross section, and to predict peptide binding affinity to major histocompatibility complex (MHC) isoforms. To enable positional interpretation of our predictions, we add indexes to the inputs to track SHAP explanations calculated from the models. Our results demonstrate that positional interpretation of models recapitulates known biochemistry and reveals new biochemistry. This positional SHAP (PoSHAP) conceptual framework provides a foothold for interpretation of other models trained from biological sequences.

**Funding:** This work was supported by the National Institutes of Health (NIH), including the National Institute of General Medical Sciences (NIGMS, https://www.nigms.nih.gov/) award number R35 GM142502 to JGM, and the National Library of Medicine (NLM, https://www.nlm.nih.gov/) training grant award number T15 LM007359 to JGM (PI: Mark Craven). The funders had no role in study design, data collection and analysis, decision to publish, or preparation of the manuscript.

**Competing interests:** The authors have declared that no competing interests exist.

This is a *PLOS Computational Biology* Methods paper.

## Introduction

Sequences are ubiquitous in biology. Nucleic acids and proteins encode information as sequences of monomeric building blocks. Sequence order is extremely important; the primary amino acid sequence of a protein uniquely determines the set of 3D structures formed after folding. Decades of effort by thousands of scientists has focused on measuring protein structures [1–3] and determining intermolecular binding [4]. Significant efforts have been devoted to protein structure prediction [5,6]. Recent advances in deep learning have achieved major milestones in protein structure prediction [7].

Deep learning is a type of machine learning that uses neural networks to learn relationships between pairs of input and output data [8]. For example, deep learning models can take inputs of peptide sequences and predict chromatographic retention time [9]. There are many types of neural network models that differ primarily in how neurons are connected. Each architecture is well suited for different types of input data. For example, convolutional neural networks (CNNs) are effective at using images as inputs [10], and recurrent neural networks (RNNs) are effective at using sequence data as input [11]. RNNs have found extensive application to natural language processing (NLP) [12], and by extension as a similar type of data, predictions from biological sequences such as peptides or nucleic acids [13]. A specific type of RNN called long short-term memory (LSTM) solves the vanishing gradient problem seen with backpropagation of RNNs [14], and thus LSTM has seen widespread use for biological sequence data.

One goal of building predictive models is to create an understandable and actionable relationship between the input and output data. Although deep learning with LSTM models is effective for making predictions from sequences, interpreting how inputs lead to specific outputs is not trivial. There are model-specific interpretation strategies, such as layer-wise relevance propagation [15] or the attention mechanism [16]. There are also strategies to enable interpretation of an arbitrary model, such as permutation importance [17], and Shapley additive explanations [18,19]. SHAP uses the game theoretic approach of Shapely values that ensures the contributions of the inputs sum to the predicted output plus a baseline [18]. SHAP is an attractive option because it can dissect interactions between inputs, for example when inputs are correlated. SHAP is also beneficial in that it can be used with any arbitrary model. However, the existing SHAP package does not directly enable sequence-dependent model interpretation.

The major histocompatibility complex (MHC) is an array of closely related genes that encode cell surface proteins that form an essential part of the adaptive immune system [20–22]. There are two main classes of MHC complexes, I and II. Peptides bound by the MHC I complex are primarily generated by the proteasome from intracellular proteins [23]. Not all degradation products are bound into the MHC class I complex, nor are all peptides bound with equal frequency. Peptides suitable for the MHC class I complex are generally between eight and ten amino acids in length, although longer peptides have been reported [24]. The sequence of the peptide is the primary determinant of binding affinity to each MHC class I complex allele. Given the polymorphism of MHC class I alleles in the human population, abundance of potential binding peptides, and the low throughput of many binding assays, the direct testing of most peptides is infeasible. Therefore, the prediction of binding affinities

through methods such as machine learning or molecular modelling could lead to improved development of vaccines against disease like cancer [25].

Extensive efforts have focused on prediction of peptide-MHC interactions [26,27]. Both classification and regression models are used to learn which peptides bind to each MHC allele, for example see O'Donnell *et al.*, Zeng and Gifford, and Liu *et al.* [28–30] However, because many reports forgo model interpretation, the learned biochemical relationships remain unknown. Other works determine relationships learned by their model, for instance both Jin *et al.* [31], and Hu *et al.* [32] used CNNs with an attention mechanism to determine the weights of the inputs on the final prediction.

Attention mechanisms have been successful in recapitulating experimentally defined binding motifs, but require that the model be constructed with attention layers. This may limit the flexibility of model architecture when designing new models. For example, attention mechanisms are specific to neural networks. Simpler models, such as random forests and XGboost, may also be more suitable for some applications, and these cannot utilize attention. Also, while attention mechanisms are currently very effective, there is always a possibility that new architectures will emerge that make interpretations using attention infeasible. Beyond this, attention is a metric of the model itself, while SHAP values are calculated on a per input basis. By looking at the model through the lens of the inputs, we can understand the model's "reasoning" behind any peptide's prediction. Attention mechanisms also do not enable dissection of interpositional dependencies between amino acids. Thus, new methods for model agnostic interpretation are desirable.

In addition to predicting binding affinities of peptides, deep learning is useful for predicting peptide properties for proteomics applications [33] including: fragmentation patterns during tandem mass spectrometry [34–36], liquid chromatographic retention time [35,37,38], and ion mobility [39]. However, attempts at model interpretation are uncommon in this body of literature. One recent paper [39] utilized SHAP to better understand the mechanics behind the collisional cross section (CCS) of peptides, but insight was limited to aggregate amino acid contributions without position context. Further work is needed to allow model-agnostic interpretation of neural networks trained from biological sequences to understand general patterns in the chemistry of peptides.

Many effective deep learning model architectures are available for making predictions from inputs of biological sequences, and there is currently no single correct choice. CNN models such as MHCflurry 2.0 [40] and LSTM models are effective at predicting MHC binding of peptides [41]. Even "simpler" models, such as random forests, have been used to predict MHC binding [42,43]. Prediction of other peptide properties like tandem mass spectra are often done with CNN or LSTM models [33]. More recently, given the extraordinary performance of transformer models like BERT [44] and GPT-3 [45] for NLP, there is an interest in transformer models for biological sequences [46].

Here we demonstrate that LSTM models easily learn to perform regression directly from peptide sequence to that peptide's properties, including affinity to various MHC alleles [32,47] and CCS [39]. Our main contribution is a strategy to interpret such models that we term "positional SHAP" (PoSHAP). Unlike other strategies that adapt the SHAP explainer [48] or another approach that enables visualization of SHAP interpretations from sequences [49], PoSHAP simply adds indexes to inputs and maintains positional context after SHAP kernelExplainer to reveal how each amino acid contributes to predicted properties. We show how this enables new analysis for specific peptides and generally across all peptide predictions. We extend the strategy to track interpositional dependence of amino acids in peptides when predicting their MHC I binding or CCS. This work therefore describes a general, broadly applicable framework for understanding notoriously abstruse deep learning models trained from biological sequences.

## Methods

### Data

Data used for training and testing the Mamu model was obtained from Haj *et al* 2020 [47], where all possible 8-, 9-, and 10-mer peptides from 82 SIV and SHIV strains were measured by fluorescent peptide array. The data consists of 61,066 entries containing the peptide sequence, peptide length, and five intensity values corresponding to the intensity obtained from the fluorescence assay for each of the five Mamu alleles tested (A001, A002, A008, B008, and B017). From the methods of Haj *et al.*, each intensity is the base 2 logarithm of the median intensity value of five replicates reported for each peptide as measured by an MS200 Scanner at a resolution of 2μm and a wavelength of 635nm [47]. For training and testing of the model, the dataset was randomly split into three categories (**S1 Fig**). Because the dataset contains truncated forms of each core peptide sequence as 8-, 9-, and 10-mers, the data splitting grouped each core sequence into unique indexes and split based on those indexes. This core sequence-based splitting ensured that training and testing data would not have truncated versions of the same peptide. The training data had 43,923 entries (71.93% of all data). The validation data to assess overfitting during training had 10,973 entries (17.97% of all data). The test data to test the overall model performance had 6,170 entries (10.10% of all data).

Data for the human MHC allele was obtained from Hu *et al.* 2019 [32], and is a compilation of data from the IEDB MHC class I binding affinity dataset (Kim *et al.*, 2014 [50], Vita *et al.*, 2015 [51], and Pearson *et al.* 2016 [52]). This dataset consists of species, allele, peptide length, peptide sequence, and a binding affinity measurement as IC50. For A*11:01, a subset of the data was chosen by selecting only the peptides between eight and ten amino acids in length with binding data for the allele. The IC50 were transformed as described in Hu *et al.* 2019 [32] and Nielsen *et al.* 2007 [53] where score = 1-log(affinity)/log(50000). Data splitting into training, validation, and test data was performed as above, split by core peptide sequences (**S2 Fig**). The training data had 4,522 entries (71.97% of all data), the validation data consisted of 1,132 entries (18.02% of all data), and the test data consisted of 629 entries (10.01% of all data).

Data for the CCS was obtained from Meier *et al.* 2021 [39]. The dataset consists of peptide sequences, peptide lengths, peptide modifications, retention times, and calculated CCS, among other values, for about 2,000,000 peptides. From this dataset, we removed all peptides that had any modifications, and for simplicity, kept only peptides with lengths of 8, 9, or 10 amino acids. The mean of the CCS were taken for remaining peptides that had the same sequence. The final dataset consisted of 45,990 entries. The data was split into training, validation, and test sets, split by core peptide sequences, as described above (**S2 Fig**). The training data consisted of 33,134 entries (72.04% of all data). The validation data consisted of 8,256 entries (17.95% of all data). The test data consisted of 4,600 entries (10.00% of all data).

### Model architecture

The Keras(2.3.0-tf) [54] interface for Tensorflow(2.2.0) [55] was used to build and train the LSTM models (**S3 Fig**). Peptide sequences were converted to integers ranging from 0 to 20 where each integer corresponds to an amino acid or the special token "END", which is used to pad peptides with length 8 or 9 to have length 10. The embedding layer takes these ten integer inputs corresponding to each position of the peptide. Each input is transformed by the embedding layer to a 10x50 dimensional matrix that is sent to the first LSTM layer [14]. The LSTM layer outputs a 10x128 dimensional matrix to a dropout layer where a proportion of values are randomly "dropped", or set to 0. For the MHC models, a second LSTM layer outputs a tensor with length 128 to a second dropout layer. Then in all models, a dense layer reduces the data dimensionality

to 64. For the MHC models, the data is then passed through a leaky rectified linear unit (LeakyRe-LU) activation before a final dropout layer, present in all models. The final dense layer produces either one or five outputs, which are trained to predict the output values (intensity, IC50, CCS). The model is compiled with the Adam optimizer [56] and uses mean squared error (MSE) loss.

## Hyperparameter search

For the Mamu MHC alleles, dropout, batch size, and the number of epochs were optimized using the hyperas wrapper for hyperopt [57]. The hyperparameter search allowed a uniform range between 0 and 0.6 for each of the three dropout layers. The search for epoch and batch size hyperparameters had binary choices. Epochs were either 1,000 or 2,000. Batch size was either 2,500 or 5,000. To ensure that our unorthodox batch sizes were acceptable, we performed the hyperparameter search again, with options of 32, 64,128, or 5,000, and a batch size of 5,000 was again selected as optimal.

For the human MHC allele model, dropout, batch size, learning rate, and the number of epochs were optimized using the hyperas wrapper for hyperopt [57]. The hyperparameter search used a uniform range between 0 and 0.8 for each of the three dropouts. The search for epochs, batch size, and learning rate hyperparameters had defined choices. Epochs were 100, 500, or 1,000. Batch size was 32, 64, 128, or 5000. Learning rate was a choice between 0.001, 0.005, 0.01, 0.05, and 0.1. To reduce overfitting, the number of epochs was fixed at 200.

For the CCS model, dropout, batch size, learning rate, and the total number of epochs were optimized using the hyperas wrapper for hyperopt [57]. The hyperparameter search for the dropout values in the two dropout layers randomly chose values from a uniform distribution between 0 and 0.8. Learning rate was a choice of 0.001, 0.005, 0.01, 0.05, and 0.1. Batch size was a choice of 32, 64, 128, and 256. Epochs were a choice of 100, 500, and 1000. To reduce overfitting, the number of epochs was fixed at 200.

For each dataset, the hyperparameter search was run with the tree of parzenestimators algorithm [58] allowing a maximum of 100 evaluations. The optimal parameters from this search are in **S3 Fig**.

## Final model training

For each dataset, the final models were re-trained using the best hyperparameters (**S3 Fig**). Loss (as MSE) for training and validation data was plotted against the training epochs to monitor overfitting (**S4 Fig**).

## Model Performance—Regression metrics

Test peptides were input to the final trained model and the predicted outputs were compared with the experimental data. Correlations between true and predicted values were assessed by MSE, Spearman's rank correlation coefficient (Spearman's $\rho$), and the correlation p-value.

## Positional SHAP (PoSHAP)

Shapely Additive Explanations (SHAP) [18] were used to determine the contribution of each position on each peptide to the peptide's overall predicted value. As the baseline, training peptide sequence data was summarized as 100 weighted samples using the SHAP kmeans method. The summarized data, the test peptide sequence data, and the trained model were input into SHAP's KernelExplainer method. The contribution of each amino acid at each position was stored in an array. The mean SHAP value of each amino acid at each position was calculated for each input dataset. Exemplary plots of the top predicted peptides were generated using

SHAP's force_plot method indexed with peptides and position [19]. Dependence plots were generated using SHAP's dependence_plot method and modified with MatPlotLib [59].

## PoSHAP compared to summary statistics

Heatmaps were created for each of the three training datasets, with the count of each amino acid at each position. An array was created with a value to represent each amino acid at each position. Each peptide was iterated over, and the value in the heatmap for each amino acid position was incremented to get the counts. To determine the top valued peptides for each dataset, the dataset was sorted by the experimental values of the training data and each peptide given a rank index. A linear regression was calculated between the rank index and experimental values of the training data. All values that were greater than the linear regression at the particular index and greater than the overall mean of the dataset were considered to be top valued peptides. The subset of top valued peptides were then arranged into an array as above to create a heatmap.

## Dependence analysis

To generate the dependence analysis tables, for each amino acid at each position, the SHAP values were split into two sets. Given each amino acid, the first set consists of the SHAP values where a specified position contains a specific amino acid. The second set consists of the remaining SHAP values for the amino acid at the position. For example, set group one to all SHAP values for a glycine in position 2 that are followed by a lysine in position 3, and compare that with group 2, which is all SHAP values for glycine in position 2 with any other amino acid in position 3. For each position and amino acid, all sets of positions and amino acids are compared. The two sets are not normally distributed and were therefore compared with a Wilcoxon Rank Sum test (also known as Mann-Whitney U-test), and the p-values are adjusted with the Bonferroni correction.

To analyze the interdependent interactions between positions and amino acids, the subset of all significant (Bonferroni adjusted P-value < 0.05) interactions were taken from the CCS dependence analysis tables. Interactions involving the "End" token were removed. The remaining interactions were grouped by distance or by expected interaction type. Interactions grouped by distance, were further grouped into either neighboring (distance = 1), near (distance = 2,3,4,5,6), or far (distance = 7,8,9). Each amino acid was grouped into the following categories: "Positive" for arginine, histidine, and lysine; "Negative" for aspartic acid and glutamic acid; "Polar" for serine, threonine, asparagine, and glutamine; "Hydrophobic" for alanine, valine, isoleucine, leucine, methionine, phenylalanine, tyrosine, tryptophan, cysteine, glycine, and proline; and "End" for interactions involving the end input. Expected interaction type was determined by the following: "Charge Attraction" by interactions between "Positive" and "Negative" categories. "Charge Repulsion" by interaction between "Positive" and "Positive" or "Negative" and "Negative" categories. "Polar" by interactions between "Polar" and "Polar," "Polar" and "Negative," or "Polar" and "Positive" categories. "Other" by interactions not noted here, including interactions such as "Polar" and "Hydrophobic." As there were very few hydrophobic interactions i.e. hydrophobic and hydrophobic, they were included with "Other." ANOVA with Tukey's post hoc test was calculated amongst the distance groups and amino acids categories to determine significance. Finally, each amino acid category was split into the distance of interaction as above, neighboring (distance = 1), near (distance = 2,3,4,5,6), or far (distance = 7,8,9). ANOVA with Tukey's post hoc test was calculated amongst the combined categories to determine significant difference.

## Additional model testing

ExtraTreesRegressor from scikit_learn [60] and XGBRegressor from xgboost [61] were used to train models from the training data for each of the three datasets. SHAP values were calculated

for the testing data using KernelExplainer with the training data summarized by SHAP's kmeans method to 100 points. SHAP values were processed with PoSHAP as above.

Three additional LSTM models were trained from the training datasets with different hyperparameters and using RMSprop as the optimizer. The model architectures remained the same. For the Mamu model, the dropout rates were 0.4839, 0.1829, and 0.1177 for dropout layers one, two, and three respectively. It had a batch size of 32 and ran for 200 epochs. Learning rate was set at the default value. For the A*11:01 model, the dropout rates were 0.0500, 0.2953, and 0.3258 for dropout layers one, two, and three respectively. It had a batch size of 32 and ran for 200 epochs. Learning rate was set at 0.01. For the CCS model, the dropout rates were 0.5324 and 0.0865 for dropout layers one and two respectively. It had a batch size of 128 and ran for 100 epochs. Learning rate was set at the default value. SHAP values were calculated for the testing data using KernelExplainer with the training data summarized by SHAP's kmeans method to 100 points. SHAP values were processed with PoSHAP as above.

## Results

### Model training and prediction

Datasets from each source consisted of a peptide sequence and a corresponding measurement, including fluorescent intensity [47], IC50 [32,50–52], or CCS [39]. For the Mamu dataset, each peptide in the table had values for five Mamu MHC class I alleles: A001, A002, A008, B008, and B017. For the human MHC and peptide ion mobility datasets, each peptide had a single value, representing IC50 and CCS, respectively. Data was split into training, validation and test sets in a manner that ensures truncated versions of the same core peptide are in the same set (**S1** and **S2** **Figs**). The LSTM models used peptide sequences converted to integers as input to an embedding layer, and learned to perform either single-output regression, for the human MHC and CCS, or multi-output regression for the outputs of the five Mamu MHC alleles. (**Figs 1** and S3**).

Despite the limited sizes of the training sets, the LSTM models achieved excellent performance on these regression tasks as evidenced by scatterplots of true values versus model predictions for the held-out test set (**Fig 2**). Training versus validation loss for the final multi-output model (**S4 Fig**) demonstrates some overfitting but not to the detriment of the model's generalizability. To prevent overfitting of the single output regression models, epochs were limited to 200. All correlations between true and predicted values were significant with p-values less than 1.0E-145.

**Positional SHAP (PoSHAP).** PoSHAP utilizes the standard SHAP package but adapts the analysis by simply appending an index to each input and maintaining positional information after the kernelExplainer interpretation, which enables tracking of each input postion's contribution to an output prediction (**S5 Fig**).

PoSHAP analysis revealed expected patterns of positional effects for experimentally supported interactions. For the Mamu allele A001, we found patterns similar to a prior publication that determined specificity experimentally with a library of peptides with single amino acid substitutions [62]. This previous study determined a preference for "...S or T in position 2, P in position 3, and hydrophobic or aromatic residues at the C terminus". Our heatmap shows a similar preference (**Fig 3**), but we also note that F/I/L is preferred at position 1, and a proline at one of the positions between 2–5. The preference for a hydrophobic amino acid in position 1 was also seen using a substitution array in the original publication of the peptide array data used to train our models [47].

For the human MHC allele A*11:01 model, PoSHAP analysis recapitulates positional relationships found through attention mechanism based models (**Fig 4A**) [32]. This pattern is in

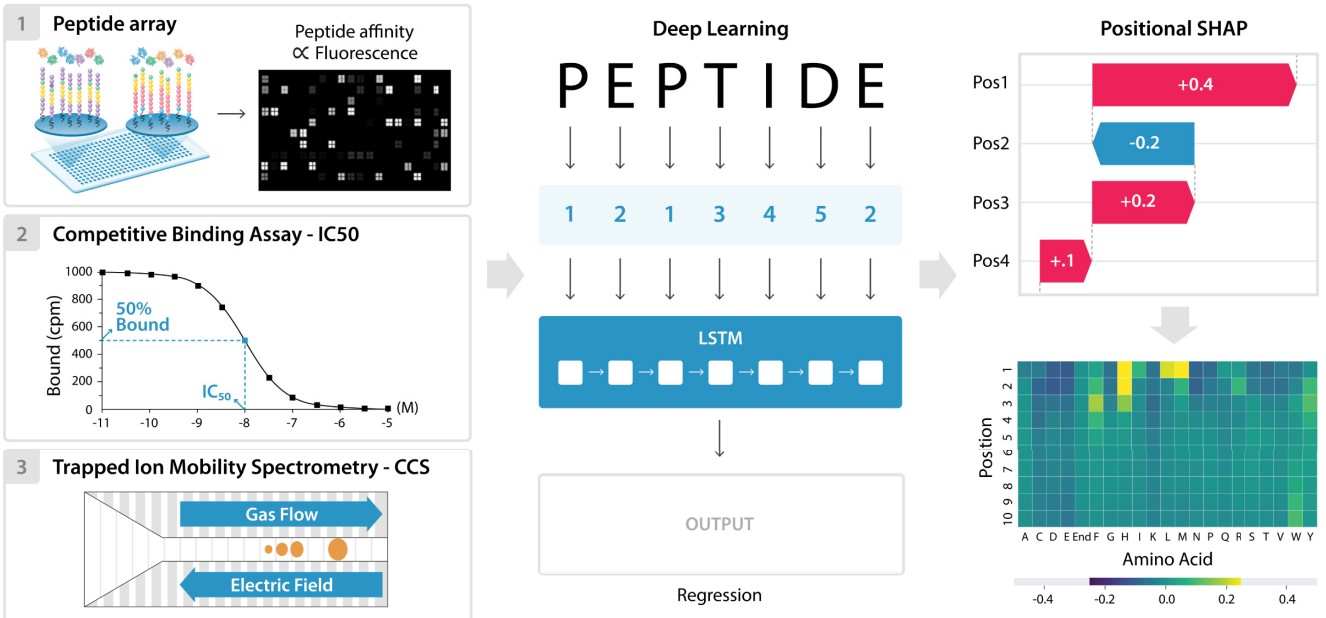

**Fig 1. Overview of data, modeling, and positional SHAP analysis for model interpretation.** Peptide sequence and output data was downloaded from Haj et al. 2020, Hu et al. 2019, and Meier et al. 2021, and used as an input for three separate deep learning models. The peptide sequences were numerically encoded, split to positional inputs, and Long Short-Term Memory (LSTM) models were trained to predict each of the outputs. These outputs included the five peptide array intensities for the Mamu MHC allele data, IC50 binding data for the human MHC A*11:01 data, and CCS for the mass spectrometry data. The trained models were then used to make predictions on a separate test subset for each of the datasets. Finally, the model interpretation method SHAP was adapted to enable determination of each amino acid position's contribution to the final prediction. This PoSHAP analysis was visualized by plotting the mean SHAP value of each amino acid at each position as a heatmap.

congruence with the experimental data for the binding of A*11:01 [63]. Jin*et al.* [31] reported anchor sites for MHC alleles from attention-based models. PoSHAP analysis matched these anchor sites based on the PoSHAP heatmap (**Fig 4A**) and the range of the SHAP values per position (**S6 Fig**). Remarkably, this was achieved for A*11:01 using a total dataset of only 4,522 examples, which shows that PoSHAP can be effective with even less than 10,000 training examples.

For the CCS model, PoSHAP analysis reflects results from experimental positional analysis performed in Meier *et al.* [39], which demonstrated the importance of the histidine residue position relative to the peptide's C- and N-terminus. Our PoSHAP analysis also shows the importance of histidine, with the highest PoSHAP values at the peptide's N-terminus, reflecting that peptides with n-terminal histidine have higher CCS. Meier *et al.* [39] also performed SHAP analysis on their own model that illustrates the contribution of each amino acid across all positions. They noticed that lysine, arginine, and histidine had the highest range of SHAP values, and suggested that this variation indicated the exact positions of these residues would influence the CCS. PoSHAP analysis agreed with this and showed that the amino acids with the greatest ranges (**S7 Fig**) also had the highest levels of positional dependence, with histidine, lysine, and arginine having the greatest overall ranges and the greatest dependence on position (**Fig 4B**).

Given the accuracy of PoSHAP in recapturing experimentally verified positional effects, it's use has promise in generating hypotheses about the analyzed systems. One example is with the CCS data. The PoSHAP analysis revealed that the three amino acids (H, K, R) that contribute the highest proportion to CCS when at the termini are all positively charged under physiological and mass spectrometry electrospray conditions. The contribution to CCS by the positively

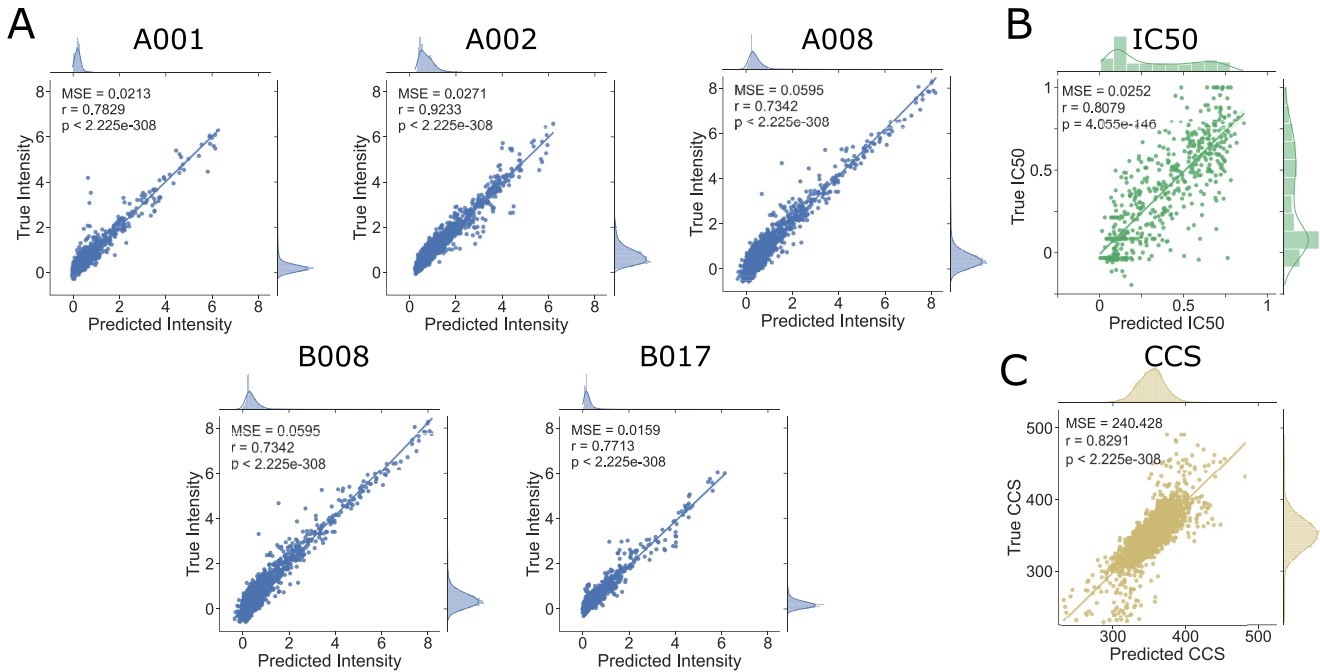

**Fig 2. LSTM Model Performance.** Held-out test peptides were input to the models and predictions were plotted against true experimental values. (A) For the Mamu allele multi-output regression model, predicted and experimental intensities were compared. (B) For the A*11:01 model, predicted and experimental IC50s were compared. (C) For the collisional cross section model, predicted and experimental collisional cross sections were compared. For each model, predicted and experimental values were compared with the Spearman's rank correlation and all demonstrated a significant (p-value < 1E-145) positive correlation (rho>0.6).

charged amino acids may be due to charge repulsion between the positive charges on adjacent amino acids in the peptide, as the vast majority of peptides used to train the model had a greater charge than +1. When the amino acids are at the termini, they have the greatest freedom to extend away from the rest of the peptide, increasing the CCS. The opposite trend is seen for the negatively charged amino acids, aspartic acid and glutamic acid, which had a slight negative effect across PoSHAP with the greatest effects at the termini.

Another application of PoSHAP is to be able to make hypotheses about the binding characteristics of the uncharacterized MHC alleles (**Fig 3**). We found that the model predicts that Mamu A001 prefers F, I, L,S, T, V, or Y in the first position, with a strong preference for S, T, or P in the second position; S and T are very similar chemically, with small, polar side chains containing hydroxyl functional groups. The heatmap of SHAP values also showed that A001 had a preference for proline between positions two and six. The preference of Mamu A002 was similar to A001 in that n-terminal serine or threonine resulted in high binding, but the preference for proline was absent. The preference map of Mamu A008 showed an opposite trend, where only the preference for early proline between positions one and four is readily apparent and the contribution of S or T is absent. Mamu B008 appears to be most selective for the amino acids near the N-terminus, with a strong preference for arginine or methionine and strong negative SHAP values for many amino acids. Finally, B017 showed a preference for L, M, or H followed by F near the N-terminus. The heatmap of SHAP values for B017 also showed a positive contribution to binding from tryptophan near the C-terminus, suggesting that the entire peptide length may play a bigger role in binding to the B017 MHC protein.

PoSHAP analysis also reveals the amino acids at each position that decrease peptide binding. All MHC alleles except for A002, and most pronounced in B008, have a strong negative

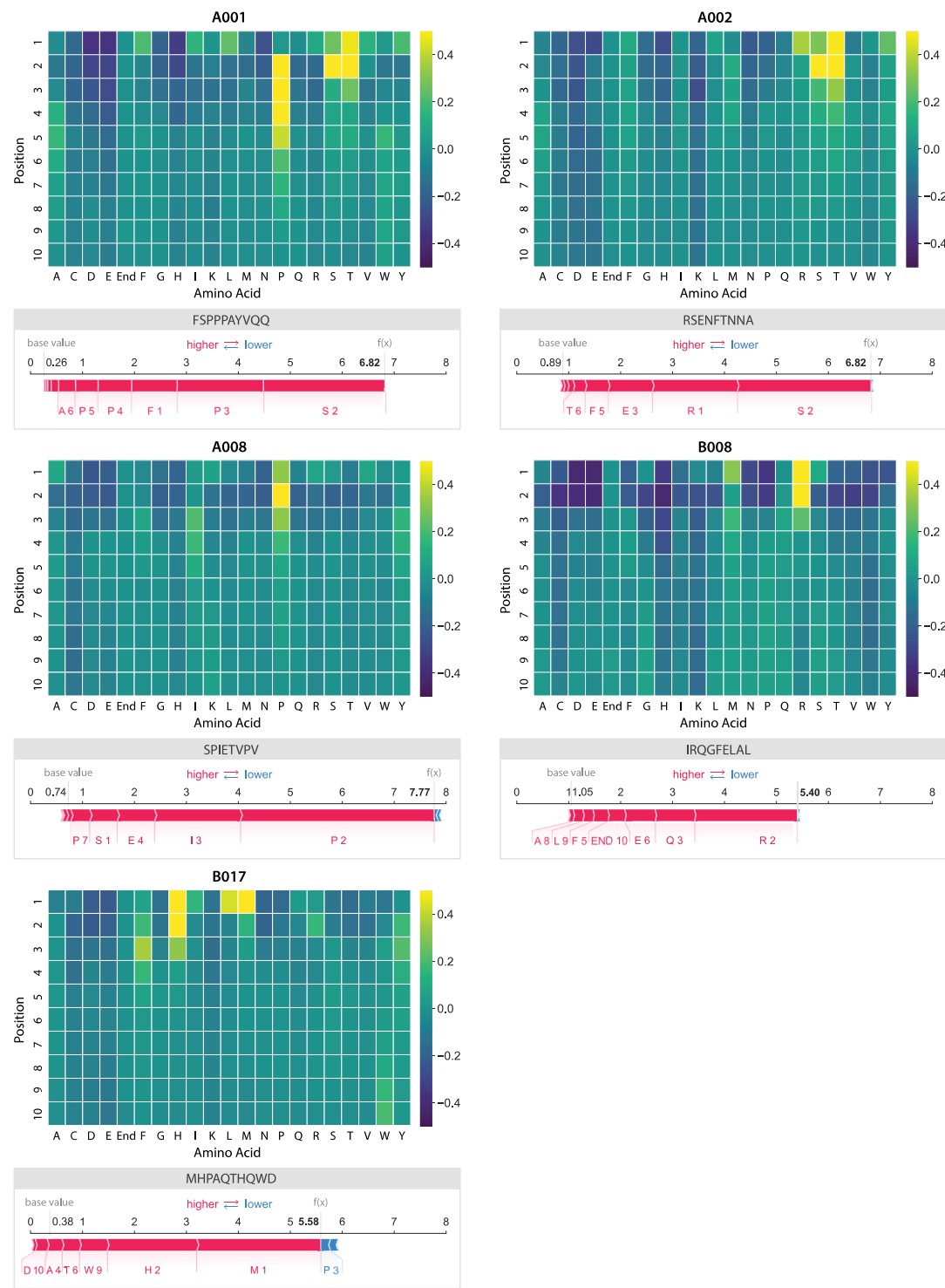

**Fig 3. Heatmaps showing PoSHAP analysis to determine amino acid binding motifs from deep learning models.** The mean SHAP values for each amino acid at each position across all peptides in the test set were arranged into a heatmap. The position in each peptide is along the y-axis and the amino acid is given along the x-axis. "End" is used in positions 9 and 10 to enable inputs of peptides with length 8, 9, or 10. For comparison, the SHAP force plot for the peptide with the highest binding prediction is shown below each allele.

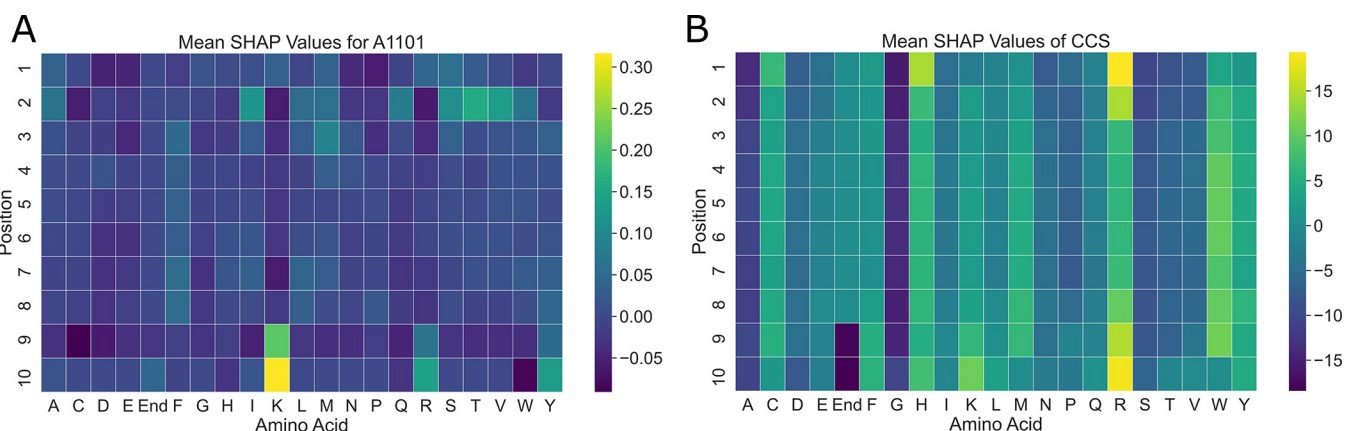

**Fig 4. PoSHAP interpretation of models trained to predict A*11:01 binding or CCS.** The mean SHAP value for each amino acid across all test peptides were calculated and arranged into heatmaps representing the values for (A) A*11:01 and (B) CCS. The position along the peptide is along the y-axis and each amino acid is listed along the x-axis.

contribution to binding prediction if there is an acidic amino acid in position one or two (i.e. D or E). For all alleles except B017, histidine near the peptide N-terminus also predicts low binding affinity.

We further show that PoSHAP can reveal the pooled binding contributions for any subsets of peptides. When the PoSHAP heatmap is filtered for the eight peptides with the highest binding predictions (top 0.013%), distinct patterns emerge (**S8 Fig**). The serine or threonine at position two remains important for the A001 and A002 alleles and can also be important to A008 binding. We also performed the same analysis with the eight peptides with the lowest binding predictions (**S8 Fig**). These PoSHAP heatmaps are primarily composed of negative SHAP values, suggesting that using this subset reveals amino acids at certain positions that are detrimental to MHC binding. Of note is the negative SHAP values for aspartic acid and glutamic acid along peptide, suggesting that positive charge may inhibit binding.

**PoSHAP versus simple summary statistics.** We wondered whether the patterns revealed by PoSHAP simply reflect the summary statistics for the high-binding or high-CCS subset of peptides. As expected, due to known differences in amino acid abundance across the proteome, the prevalence of amino acids was different across the training data and were also heterogeneous across positions (**Fig 5A**). To determine the subset of high CCS peptides, peptides were ordered in the training set by their CCS rank and then linear regression was performed to get the average trend line (**Fig 5B**). Any peptide above that trendline and the overall mean was defined as "high CCS", and the frequency of amino acids at each position in this set was summarized using a heatmap (**Fig 5C**). Compared to the statistical amino acid frequencies, PoSHAP suggests a greater importance of arginine at both termini, the importance of tryptophan to increase CCS becomes apparent, and interior glutamic acid contributes less to high CCS than the frequencies would suggest (**Fig 5D**). The same analysis was repeated for MHC data (**S9** and **S10** Figs). This demonstrates that PoSHAP finds non-linear relationships between the inputs and the outputs that are not present by simple correlation.

## PoSHAP Interpositional dependence analysis

The SHAP value of any position is dependent on the values of all other positions in the peptide. PoSHAP values for each amino acid at each position were split based off of the amino acid at another position across all peptides. This enabled the determination of the dependence of a PoSHAP value on the presence of an amino acid at any and all other positions. This method

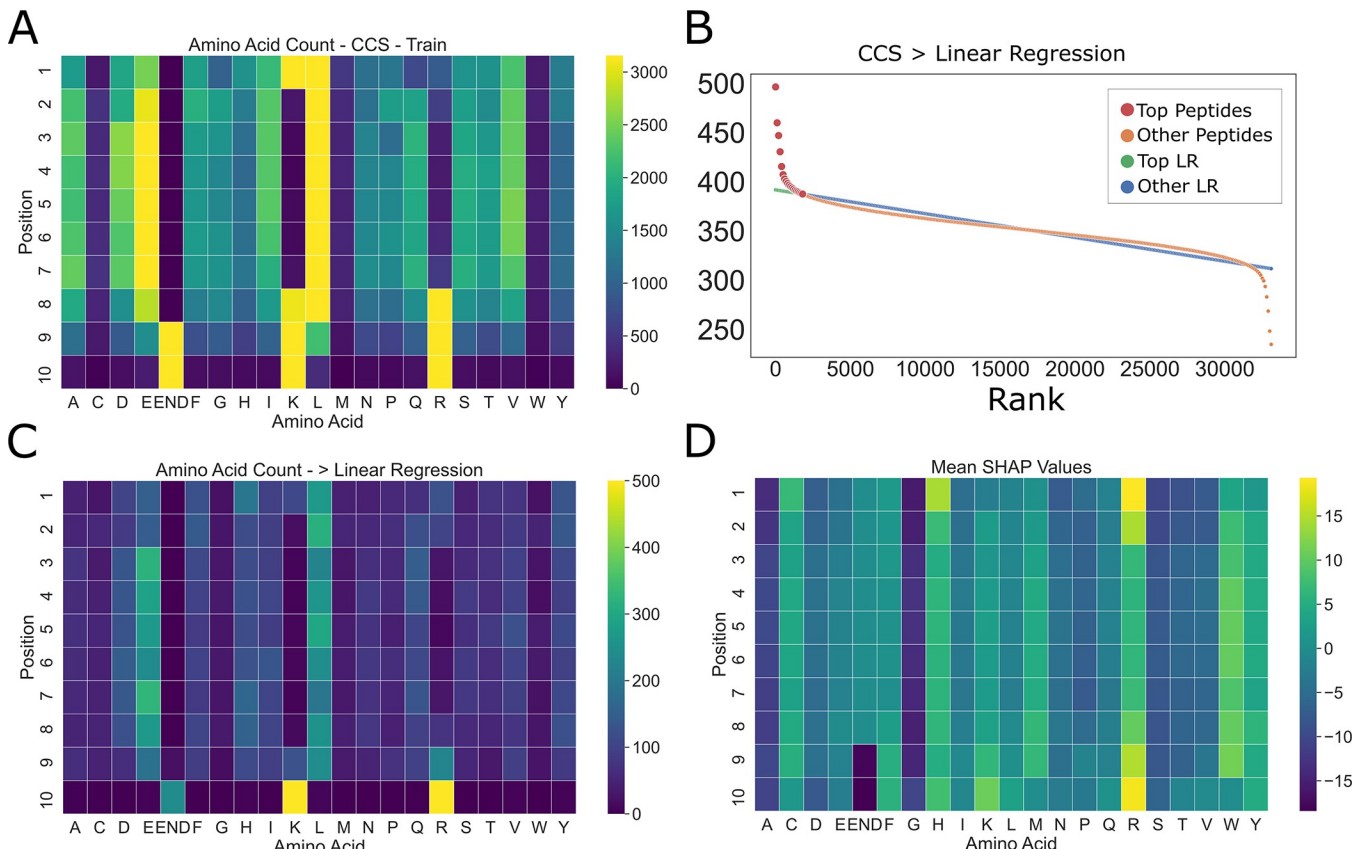

**Fig 5. Amino acid summary statistics differ from PoSHAP values for the CCS data.** (A) Amino acid counts as a function of position for training data. (B) Procedure for picking the 'top peptides' with the highest CCS. Linear regression was performed on the peptides ranked by their actual CCS value. Any peptide that fell above the trendline and overall mean were defined as 'top peptides'. (C) Counts of amino acids for the top peptides were summarized in a heatmap. (D) Mean SHAP values across amino acids and positions from PoSHAP analysis.

also enabled determination of the significance and magnitude of the dependence by comparing the means and calculating a Wilcoxon Rank Sum test with a Bonferroni correction (**S1, S2 and S3** Tables). The original SHAP package provides a means to illustrate these relationships through its dependence plots.

Using the dependence analysis, we were able to discover significant positional relationships for each of the three models we trained. For the Mamu alleles, the most striking relationship was observed with the model trained on the A001 dataset (**Fig 6**). As previously mentioned, it can be seen in the heatmaps that the highest SHAP values are observed for serine and threonine near the N-terminus of the peptide (**Fig 3**). However, the top predicted peptides do not show the same pattern, instead beginning with either a phenylalanine or a leucine, and continuing with a serine or threonine (**S8 Fig**). Among the calculated interpositional relationships with the greatest significance and magnitude are between the leucine and threonine (**S1 Table**, Bonferroni adj. p-value 1.72E-22) and the phenylalanine and serine (**S1 Table**, Bonferroni adj. p-value6.44E-8) between the first and second positions (**Fig 6**). Additionally, threonine or serine followed by a proline between positions one and two (**S1 Table**, Bonferroni adj. p-values 3.61E-6, 6.62E-8, respectively), or two and three (Bonferroni adj. p-values 1.16E-7, 4.24E-5, respectively) were significant. This suggests that the most important motif for binding is Thr-Pro or Ser-Pro and that the ideal binding motif for A001 is Leu-Thr-Pro or Phe-Ser-Pro at the N-terminus of the peptide.

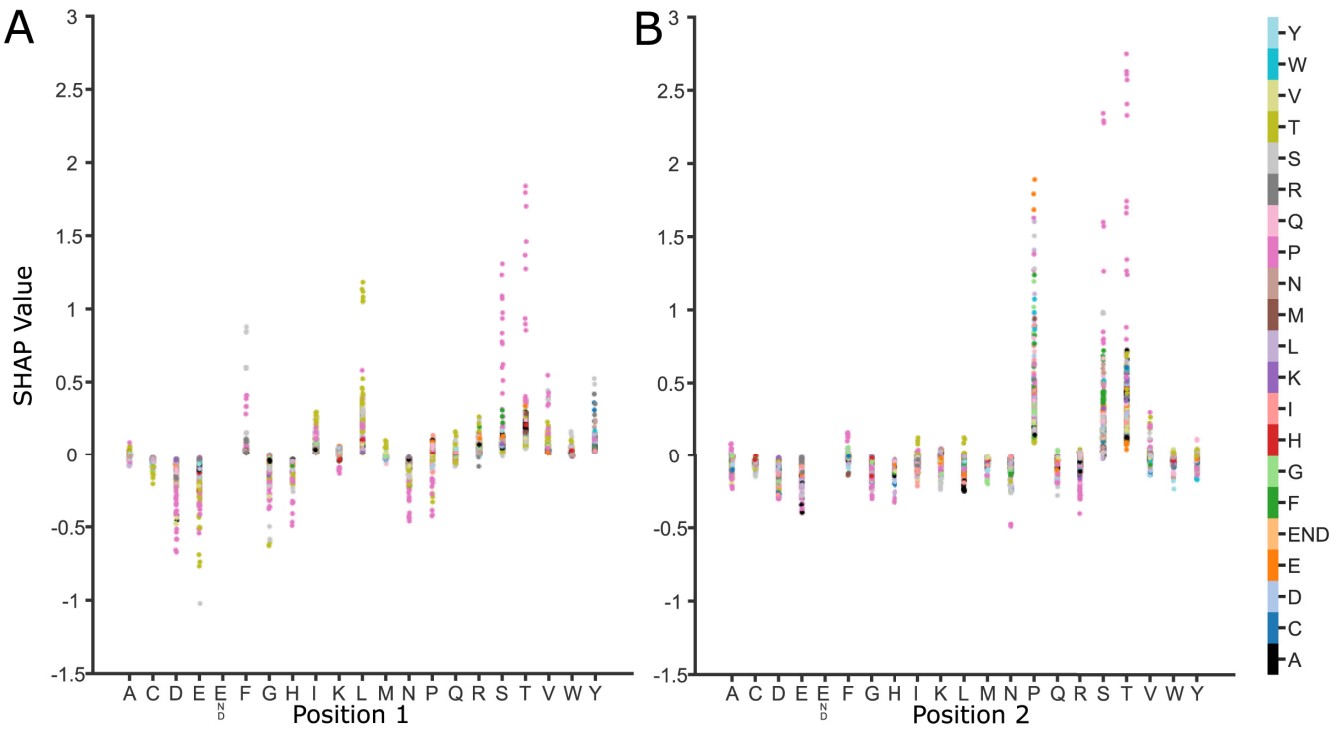

**Fig 6. SHAP dependence plots for allele Mamu-A001 show how relationships between sequential amino acids contribute to binding.** Each graph represents a pair of positions in the peptide, here (A) positions one and two and (B) positions two and three. The x-axis lists each possible amino acid for that position and the y-axis shows the SHAP value. Each point represents a peptide with the listed amino acid at that position on the x-axis and the amino acid in the subsequent position is shown by color. This shows how the range of SHAP values for a particular amino acid at a specific location is reflective of the dependence of other amino acid positions.

For the human allele A\*11:01 model, there were very few significant interactions, potentially due to the smaller size of the dataset. However, there were still a couple notable interactions. These are those between lysine at the ninth position and serine or leucine at position two (**S2 Table**, Bonferroni adj. p-values 0.001, 1.43E-5, respectively). Both of these have significantly greater SHAP contributions when lysine is at position nine. This may reflect that there is some flexibility with the earlier root site when lysine is bound and may demonstrate that the model had learned the length of the binding motif between the second position and the C-terminus (31) (**S2 Table** and **S6B Fig**).

For the CCS model, the interpositional interactions are different as they rely on the chemical interactions within the peptide itself, rather than interacting with a binding site. That is, interactions which promote peptide compaction will reduce CCS, and those that promote extension will generally increase CCS. To further determine how PoSHAP can reveal the important amino acids and positions generally, additional metrics were derived from the subset of significant amino acid pairs from the PoSHAP dependence analysis. All significant interactions (Bonferroni adj. p-value < 0.05) from the CCS model interpretation (**S3 Table**) were used to compute the absolute magnitude of the difference in SHAP value as a function of the distance between those residues (**Fig 7**). Absolute SHAP differences between interdependent amino acids were higher when the interaction was with the neighbor amino acid (ANOVA with Tukey's posthoc test p-value = 0.0426), or distant amino acids (distance 7–9, ANOVA with Tukey's posthoc adjusted p-value = 0.001) versus intermediate amino acids (distance 2–6) (**Fig 7A**). This suggests that amino acids interact more strongly with their neighbors because the R groups are adjacent and have stronger interactions with those further along the

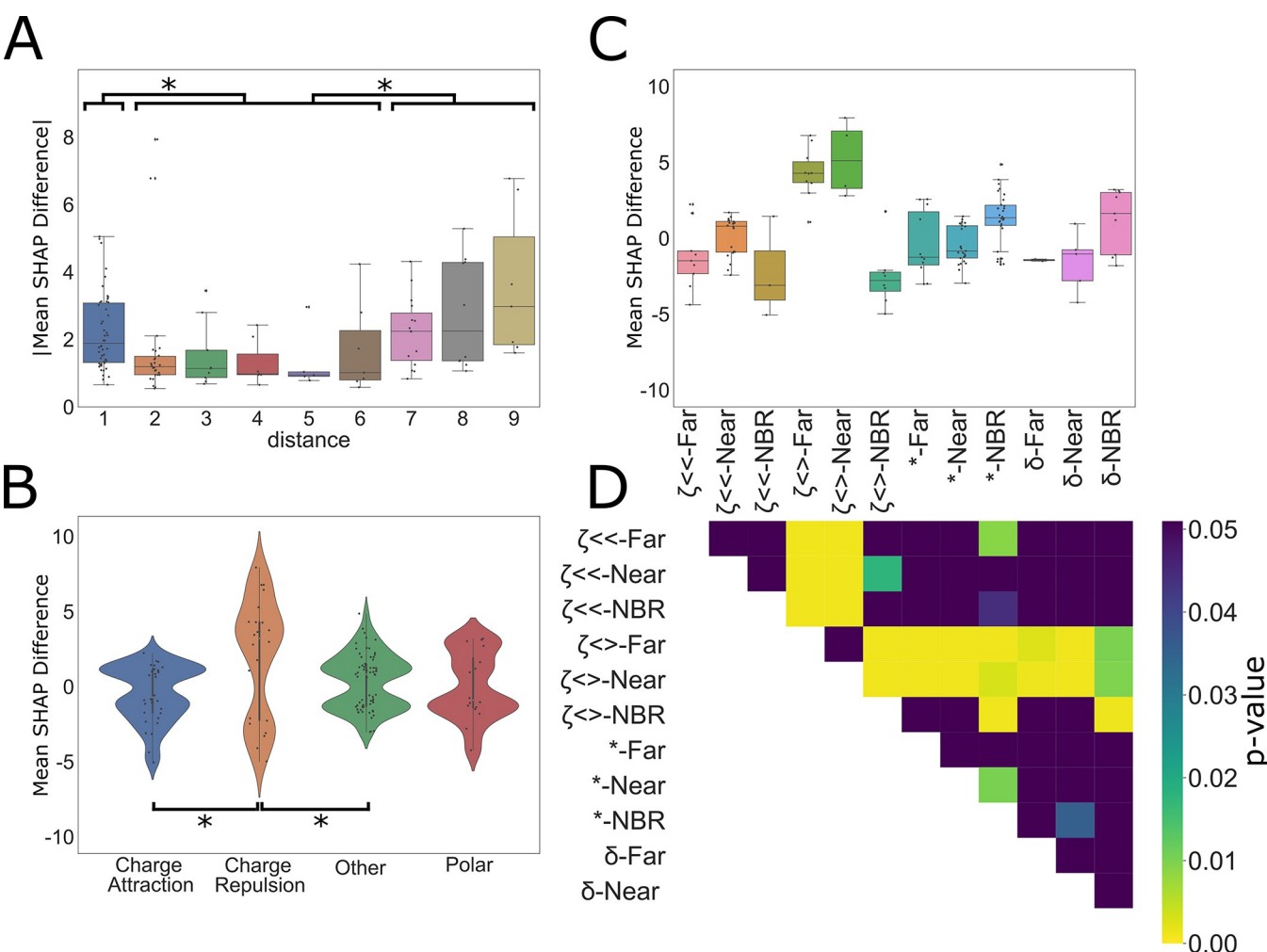

**Fig 7. Dependence analysis of CCS model.** (A) Significant (Bonferroni corr. P-value < 0.05) values were taken from the interpositional dependence analysis and the difference in the mean between the interdependent amino acids SHAP values and the remaining amino acids at each compared position pair were grouped based on the distance between the dependent interaction, (B) the category of interaction, or (C) distance and interaction category. Categories are labelled by the following for the combined bar plot and heatmap: ζ<< = charge attraction, ζ<> = charge repulsion, * = other, and δ = polar. For the distance analysis, interactions were grouped into three categories, neighboring (distance = 1), near (distance = 2, 3, 4, 5,6), and far (distance = 7, 8, 9). * indicates significance (ANOVA with Tukey's post hoc test p-value < 0.05). For the interaction categories in (B) and (C), each interaction was grouped by the expected type of interaction between the two amino acids. Significant differences between interaction types are noted by the pairing by lines (ANOVA with Tukey's post hoc test p-value < 0.05). (D) Significant differences between combined categories are illustrated by the heatmap where significant values (ANOVA with Tukey's post hoc test p-value < 0.05) are designated by colors other than purple. Exact p-values for each are provided in S4 Table. repulsive molecular interactions, including charge repulsion and "other" interactions (likely steric interactions or interactions between the termini) increased predicted CCS. Notably, there were very few significant hydrophobic interactions. This may reflect that hydrophobic interactions between amino acids in a peptide act to minimize contact with a polar solvent, rather than acting as an attractive force itself. Peptides lose polar solvent (water) during the electrospray process, which may prevent significant hydrophobic interactions, which might contradict prior work [64].

peptide because of the flexibility of the chain, but interactions at an intermediate distance have a lesser magnitude of effect.

Additionally, there are interesting differences in the interactions of the amino acid among the significant set of interactions (**Fig 7B**). All significant interactions from the CCS data (**S3 Table**, Bonferroni adj. p-value < 0.05) within the peptides were grouped by the expected interaction type occurring between the amino acids based off of the chemistry of their side chains. Interestingly, this analysis revealed that generally attractive molecular interactions, including charge attraction and polar interactions on average decreased predicted CCS while generally

Though it is evident that the mean of each interaction type corresponds to the expected impact those interactions would have on CCS, each of the interaction dependence plots are bimodal, with some interactions increasing CCS and some decreasing it. To dissect this observation further, we combined the two methods of splitting the data to see if the bimodality of interaction types would be resolved by distance (**Fig 7C**). Though definitive conclusions cannot be made for most categories, likely due to the ever decreasing sample size by splitting, of note is the difference between neighboring charge repulsion and non-neighboring charge repulsion. Neighboring charge repulsion seems to decrease CCS while distant charge repulsion increases CCS (see adjusted p-value from Tukey's posthoc test in **Fig 7D**). When distant, charge repulsion makes intuitive sense as the amino acids are forced apart, linearizing the peptide and increasing the surface area. When neighboring, it is possible that the repulsion causes a kink in the linear peptide, decreasing the cross section. Overall, these analyses demonstrate that the models were able to learn fundamental chemical properties of the amino acids and through PoSHAP analysis we were able to uncover them.

Finally, to ask if the absolute positions of amino acids in the peptide are relevant for the interaction, the data was split into 8, 9, or 10mers before analysis (**S11 Fig**). This revealed that there may be interactions between the termini, but this effect may be difficult to observe because there are significantly fewer 8mers and 9mers in the CCS dataset.

## PoSHAP results are model-dependent

PoSHAP uses the SHAP KernelExplainer method, which is based on Local interpretable model-agnostic explanations (LIME). Using the general KernelExplanner method enables direct comparison of interpretations produced by different models trained from the same data. To ask whether PoSHAP interpretation changes based on the model used, the CCS data was used to train XGboost or ExtraTrees models. Surprisingly, the XGboost model performed better than the LSTM model with regard to MSE and spearman rho between true and predicted values in the test set (**Fig 8A**). ExtraTrees was slightly worse than the other two models. The model interpretation heatmaps from PoSHAP were similar between the LSTM and XGboost, but the interpretation from the ExtraTrees model was missing the high average SHAP due to n-terminal histidine or arginine (**Fig 8B**). Even though XGboost produced a similar PoSHAP heatmap, the interpositional dependence with regard to distance (**Fig 8C**) and chemical interactions (**Fig 8D**) were muted. This shows that the choice of model is important for revealing positional interactions.

Given the dependence of the model interpretation results on the model used, the same model architecture trained with different parameters might result in different model interpretation. Models for each of the three tasks mentioned here were retrained with different hyperparameters including the "RMS prop" optimizer instead of Adam. Each model produces similar or better prediction performance compared to the initial version, and the model interpretation by PoSHAP was almost identical to the previous results in all three cases (**S12**, **S13 and S14** Figs). This suggests that the model architecture drives the differences in interpretation, not the model training process.

## Discussion

Here we demonstrate the concept of PoSHAP analysis to interpret machine learning models trained from inputs of biological sequences. We show how PoSHAP can reveal amino acid motifs that influence peptide MHC I binding or CCS. We further describe how PoSHAP enables understanding of interpositional dependence of amino acids that result in high MHC I affinity. Finally, we show how PoSHAP reveals the chemical interactions within peptides that

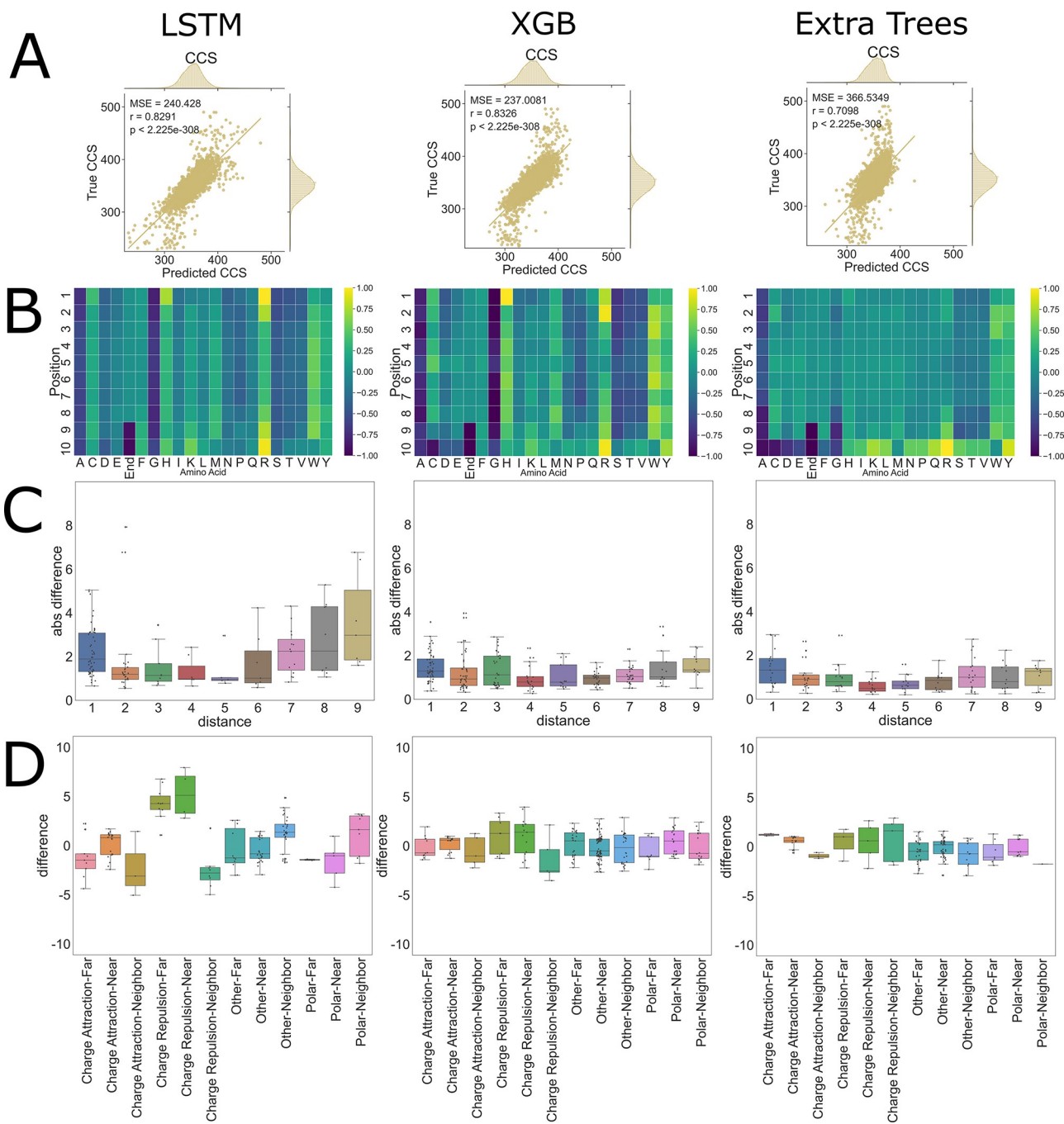

**Fig 8. CCS PoSHAP of Various Machine Learning Models.** PoSHAP analysis was performed on two additional machine learning models, Extra Trees, and Extreme Gradient Boosting (XGB). (A) Predictions were plotted against experimental values and the Mean Squared Error and r values are reported for each model. (B) PoSHAP heatmaps were created for each model, standardized by the highest value in each heatmap, illustrating an increase in model complexity as more sophisticated models are used. Dependence analysis was performed on each model and the significant interactions are plotted by (C) distance and by (D) combined distance and interaction type.

alter their CCS. Overall, the three modeling examples laid out herein serve as a tutorial for PoSHAP interpretation of almost any model trained from almost any biological sequence.

Although there are many effective neural network models for biological sequences, there are a dearth of methods to understand those models. Thus, PoSHAP fills a gap in the biological

machine learning community. Prior studies have used sequence logos from the top predictions [65], but this method doesn't ask the model what was learned and instead is observed-sequence centric. Another approach used by DeepLigand [29] is to apply Sufficient Input Subset (SIS) analysis [66], which attempts to reduce inputs to the minimal values required for prediction. While useful in many contexts, SIS is only amenable to classification models and does not provide contribution values for each input. A third approach is to create an interpretable model through the construction of the model itself. One example of this is by using attention mechanism based models, such as what has been done with ACME [32]. However, this method of model interpretation limits the architecture of the model.

There are several benefits of PoSHAP over competing methods. First, PoSHAP determines important residues despite biases in the frequencies of amino acids (**Figs 5,** S9 **and** S10). PoSHAP is also applicable to any model trained from sequential data (**Fig 8**) and enables dissection of interpositional dependencies (**Figs 6 and** 7). Finally, we include a clearly explained jupyter notebook on Github that will take any model and dataset and perform PoSHAP analysis.

Altogether the advances described herein are likely to find widespread use for interpreting models trained from biological sequences, including models not covered here such as those to predict tandem mass spectra (reviewed in [33]).

## Supporting information

**S1 Fig. Details of the data distributions and splitting.** The data was split into three subsets. Training data made up 72% of the overall data and was used directly to train the model. Validation data made up 18% of the overall data and was used to monitor overfitting. Test data made up 10% of the total data and was used to test the final model's performance. The intensity distributions for each data subset were plotted for each allele to ensure that each maintained the same distribution. Note the difference in y-axis scales.
(EPS)

**S2 Fig. Details of the data distributions and splitting for A***11:01 **and CCS.** The data was split into three subsets. Training data made up 72% of the overall data and was used directly to train the model. Validation data made up 18% of the overall data and was used to monitor overfitting. Test data made up 10% of the total data and was used to test the final model's performance.
(EPS)

**S3 Fig. Summary of LSTM model architecture.** (A) The architecture of the model consists of an embedding layer with 10 inputs with 21 dimensions each, representing each position of the peptide and each of the numeric representations of the possible amino acids and the end marker. This is followed by a pair of LSTM and dropout layers, with the dropout ratios determined by a hyperparameter search. Following the LSTM layers are a dense layer, a leaky ReLU activation layer, a final dropout layer, and a final dense layer with five outputs, each representing the intensity of the corresponding allele. The model was trained with a batch size of 5000 for 1000 epochs. (B) Hyperopt was used to determine the ideal hyperparameters for the model using a tree of parzenestimators algorithm over 100 evaluations. * indicates a hardcoded hyperparameter.
(TIF)

**S4 Fig. Mean squared error loss over training.** The models were trained for either 1000 or 200 epochs and the loss from mean squared error between predictions and true, known values were plotted for both the training data and the validation data. For the MAMU model,

validation loss diverges from the test loss around epoch 175, indicating some amount of over-fitting, however, the MSE of both datasets continues to decrease as the model is trained over 1000 epochs. For the A*11:01 model, test and validation loss were similar, until around epoch 200 when the model was finished training. For the CCS dataset, the validation loss started and remained lower throughout the training.
(EPS)

**S5 Fig. SHAP Forceplots Demonstrating PoSHAP Indexing.** Two forceplots were created with the SHAP forceplot method of the third peptide in the CCS testing set. (A) shows the plot with encoded inputs mapped to their amino acid. (B) shows the plot with the encoded inputs mapped to their amino acid and position. The addition of positional indexing removes the ambiguity of contributions, for example, glutamine having both a positive and a negative SHAP contribution to the prediction of the third peptide.
(EPS)

**S6 Fig. SHAP value ranges by position.** SHAP values were arranged by position in the peptides and their distributions were plotted as violin plots, with the quartile ranges and total range illustrated by the box and whisker plot within each. (A) Each of the five modeled MAMU alleles, (B) human MHC A*11:01, and (C) CCS are displayed.
(EPS)

**S7 Fig. SHAP value ranges by amino acid.** SHAP values were arranged by amino acids across all positions in the peptides and their distributions were plotted as violin plots, with the quartile ranges and total range illustrated by the box and whisker plot within each. (A) Each of the five modeled MAMU alleles, (B) human MHC A*11:01, and (C) CCS are displayed. The "end" input token is represented by x.
(EPS)

**S8 Fig. Pooled PoSHAP for bottom and top predicted subsets of the data.** The mean SHAP values for each amino acid at each position were calculated for the peptides with (A) the bottom or (B) top 0.013% predicted intensity (top 8 peptides) for the "A" Mamu alleles. Due to the small sample size, most of the amino acid positions have a value of zero. The positions with extreme values, however, illustrate important amino acids for prediction. Notably for A001 and A002, aspartic acid and glutamic acid contribute to low prediction along the peptide, suggesting charge may inhibit binding. For the top predictions, phenylalanine or leucine are important at the first position for both A001 and A008. A serine or threonine at position two is important for A001, A002, and A008. All alleles demonstrate the importance of a proline near the middle of the peptide.
(EPS)

**S9 Fig. Amino acid summary statistics differ from PoSHAP values for the A001 MAMU MHC I data.** (A) Amino acid counts as a function of position for training data. (B) Procedure for picking the 'top peptides' with the highest CCS. Linear regression was performed on the peptides ranked by their actual CCS value. Any peptide that fell above the trendline and overall mean were defined as 'top peptides'. (C) Counts of amino acids for the top peptides were summarized in a heatmap. (D) Mean SHAP values across amino acids and positions from PoSHAP analysis. For the MAMU model, the amino acid frequencies of the input peptides show no obvious preference for amino acid position, but some amino acids are over-represented overall. The presence of the "end" token is more likely to be a high binder statistically (C), but the PoSHAP reveals that this end token is not the main determinant of binding (D).
(EPS)

**S10 Fig. Amino acid summary statistics differ from PoSHAP values for the human A1101 MHC I data. (A)** Amino acid counts as a function of position for training data. The distribution of amino acids in this data. **(B)** Procedure for picking the 'top peptides' with the highest CCS. Linear regression was performed on the peptides ranked by their actual CCS value. Any peptide that fell above the trendline and overall mean were defined as 'top peptides'. **(C)** Counts of amino acids for the top peptides were summarized in a heatmap. **(D)** Mean SHAP values across amino acids and positions from PoSHAP analysis. There are clear differences between the summary statistics of top peptides **(C)** and PoSHAP heatmap **(D)**. For example, the end token is prominent in the summary statistics absent from the PoSHAP interpretation. Also, the preference for S/T/V at position two is tempered according to PoSHAP, but would be determined to be very important by the summary statistics.
(EPS)

**S11 Fig. SHAP Values of Collisional Cross Section by Peptide Length.** The impact of peptide length on SHAP values was explored for the CCS data. The dataset was split into peptides of length 8, 9, and 10. All SHAP values were plotted as violin plots. The mean SHAP values were plotted in heatmaps by position and amino acid and standardized. Significant interactions by dependence analysis were plotted in bar charts by distance between interactions.
(EPS)

**S12 Fig. PoSHAP Analysis of Mamu A001 With Unoptimized Hyperparameters and RMSprop.** A new model for the Mamu data was trained using the same architectures but with different hyperparameters and RMSprop as the optimization algorithm. (A) Loss was plotted as mean squared error compared to the validation data. (B) Similar metrics for MSE, r, and p-values were obtained. (C) Similar patterns are also observed for the PoSHAP heatmap of A001. A dependence plot for A001 shows similar patterns to the Adam optimized model, including the positional dependence of proline at position two for high SHAP values of serine and threonine.
(EPS)

**S13 Fig. PoSHAP Analysis of A:11*01 With Unoptimized Hyperparameters and RMSprop.** A new model for the A:11*01 data was trained using the same architectures but with different hyperparameters and RMSprop as the optimization algorithm. (A) Loss was plotted as mean squared error compared to the validation data. (B) Similar metrics for MSE, r, and p-values were obtained. (C) Similar patterns are also observed for the PoSHAP heatmap of A:11*01. The SHAP ranges by position plot for A:11*01 shows similar patterns to the Adam optimized model, including the largest range of SHAP values at position two, nine, and ten.
(EPS)

**S14 Fig. PoSHAP Analysis of CCS With Unoptimized Hyperparameters and RMSprop.** A new model for the CCS data was trained using the same architectures but with different hyperparameters and RMSprop as the optimization algorithm. (A) Loss was plotted as mean squared error compared to the validation data. (B) Similar metrics for MSE, r, and p-values were obtained. (C) Similar patterns are also observed for the PoSHAP heatmap of CCS. Dependence analysis was performed on the dataset and the combined distance-interaction type bar plot shows similar relationships between the groupings, notably charge repulsion's split.
(EPS)

**S1 Table. Dependence Analysis of Mamu Alleles.** Dependence analysis was performed on the SHAP values for the binding of the five Mamu alleles. The Mann-Whitney test was calculated for each amino acid at each position between the SHAP values given another amino acid

at another position, and all remaining SHAP values for that amino acid at that position. P-values were Bonferroni corrected and means and differences between each set of SHAP values is reported for each.
(XLSX)

**S2 Table. Dependence Analysis of A\*11:01.** Dependence analysis was performed on the SHAP values for the binding of the A\*11:01. The Mann-Whitney test was calculated for each amino acid at each position between the SHAP values given another amino acid at another position, and all remaining SHAP values for that amino acid at that position. P-values were Bonferroni corrected and means and differences between each set of SHAP values is reported for each.
(CSV)

**S3 Table. Dependence Analysis of CCS.** Dependence analysis was performed on the SHAP values for the CCS data. The Mann-Whitney test was calculated for each amino acid at each position between the SHAP values given another amino acid at another position, and all remaining SHAP values for that amino acid at that position. P-values were Bonferroni corrected and means and differences between each set of SHAP values is reported for each. Categories of each interaction type are also listed for each interaction.
(CSV)

**S4 Table. Statistical Tests of Interaction Types.** Tukey's range test was performed on each of the interaction type comparisons. Compared categories, means, and corrected p-values are reported.
(CSV)

## Acknowledgments

This research was completed in part with computational resources and technical support provided by the Research Computing Center at the Medical College of Wisconsin. We thank Dasom Hwang for help with graphic design.

## Author Contributions

**Conceptualization:** Quinn Dickinson, Jesse G. Meyer.

**Data curation:** Quinn Dickinson, Jesse G. Meyer.

**Formal analysis:** Quinn Dickinson, Jesse G. Meyer.

**Funding acquisition:** Jesse G. Meyer.

**Investigation:** Quinn Dickinson, Jesse G. Meyer.

**Methodology:** Quinn Dickinson, Jesse G. Meyer.

**Project administration:** Jesse G. Meyer.

**Resources:** Quinn Dickinson, Jesse G. Meyer.

**Software:** Quinn Dickinson, Jesse G. Meyer.

**Supervision:** Jesse G. Meyer.

**Validation:** Quinn Dickinson, Jesse G. Meyer.

**Visualization:** Quinn Dickinson, Jesse G. Meyer.

**Writing – original draft:** Quinn Dickinson, Jesse G. Meyer.

**Writing – review & editing:** Quinn Dickinson, Jesse G. Meyer.

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
