## [Decision Letter · Decision Letter 0]

18 Nov 2021

Dear Dr. Meyer,

Thank you very much for submitting your manuscript "Positional SHAP (PoSHAP) for Interpretation of Machine Learning Models Trained from Biological Sequences" for consideration at PLOS Computational Biology. As with all papers reviewed by the journal, your manuscript was reviewed by members of the editorial board and by several independent reviewers. The reviewers appreciated the attention to an important topic. Based on the reviews, we are likely to accept this manuscript for publication, providing that you modify the manuscript according to the review recommendations.

Sincerely,

Ilya Ioshikhes

Deputy Editor

PLOS Computational Biology

Ilya Ioshikhes

Deputy Editor

PLOS Computational Biology

[LINK]

Reviewer's Responses to Questions

**Comments to the Authors:**

Reviewer #1: Mahematical notations are still missing in Positional SHAP in explanation

**Have the authors made all data and (if applicable) computational code underlying the findings in their manuscript fully available?**

Reviewer #1: None

PLOS authors have the option to publish the peer review history of their article (what does this mean?). If published, this will include your full peer review and any attached files.

Reviewer #1: No

Figure Files:

Data Requirements:

Reproducibility:

References:

---

## [Decision Letter · Decision Letter 1]

9 Dec 2021

Dear Dr. Meyer,

We are pleased to inform you that your manuscript 'Positional SHAP (PoSHAP) for Interpretation of Machine Learning Models Trained from Biological Sequences' has been provisionally accepted for publication in PLOS Computational Biology.

Best regards,

Ilya Ioshikhes

Deputy Editor

PLOS Computational Biology

Ilya Ioshikhes

Deputy Editor

PLOS Computational Biology

Reviewer's Responses to Questions

**Comments to the Authors:**

Reviewer #1: Comments are fixed

**Have the authors made all data and (if applicable) computational code underlying the findings in their manuscript fully available?**

Reviewer #1: None

PLOS authors have the option to publish the peer review history of their article (what does this mean?). If published, this will include your full peer review and any attached files.

Reviewer #1: No

---

## [Editor Report · Acceptance letter]

18 Dec 2021

PCOMPBIOL-D-21-01694R1 

Positional SHAP (PoSHAP) for Interpretation of Machine Learning Models Trained from Biological Sequences

Dear Dr Meyer,

I am pleased to inform you that your manuscript has been formally accepted for publication in PLOS Computational Biology. Your manuscript is now with our production department and you will be notified of the publication date in due course.

With kind regards,

Katalin Szabo
